

# Explainability of deep learning models in medical video analysis: a survey

Michal Kolarik, Martin Sarnovsky, Jan Paralic and Frantisek Babic

Department of Cybernetics and Artificial Intelligence, Technical University in Kosice, Kosice, Slovakia

## ABSTRACT

Deep learning methods have proven to be effective for multiple diagnostic tasks in medicine and have been performing significantly better in comparison to other traditional machine learning methods. However, the black-box nature of deep neural networks has restricted their use in real-world applications, especially in healthcare. Therefore, explainability of the machine learning models, which focuses on providing of the comprehensible explanations of model outputs, may affect the possibility of adoption of such models in clinical use. There are various studies reviewing approaches to explainability in multiple domains. This article provides a review of the current approaches and applications of explainable deep learning for a specific area of medical data analysis—medical video processing tasks. The article introduces the field of explainable AI and summarizes the most important requirements for explainability in medical applications. Subsequently, we provide an overview of existing methods, evaluation metrics and focus more on those that can be applied to analytical tasks involving the processing of video data in the medical domain. Finally we identify some of the open research issues in the analysed area.

## INTRODUCTION

Recent Artificial Intelligence (AI) systems that are based on machine learning algorithms excel in many fields. AI can outperform humans in visual tasks or strategic games, but it is also becoming an indispensable part of our everyday lives, such as online services that analyze our shopping carts or systems that allow us to make decisions based on data. AI systems based on black-box models are used in many areas today. These systems used in smartphone applications or online services do not have key requirements for model explainability but focus mainly on model accuracy and cost. If such a model fails and, *e.g.*, does not recognize the person logging into the system or the translation system makes a grammatical error in translation, it usually does not have major consequences. The requirements for transparency and trust in these applications are low. However, these requirements play an important role in applications critical to human safety. They can even be decisive when deploying such a system if the consequences of an AI decision can be life-threatening, *e.g.*, in autonomous vehicles or in the medical domain. Therefore, explainability is more important, especially in these areas, and promotes increased transparency of the model and trust in the deployed AI-based system. In order

Corresponding author
Martin Sarnovsky,
martin.sarnovsky@tuke.sk

to understand how an AI model makes predictions, we need to know how it works and based on what evidence it makes the decisions. Explainable Artificial Intelligence (XAI) methods provide tools that can help to address these issues. In addition, there are legislative requirements for clarity and transparency in the processing of personal data as well as medical data.

This article aims to provide an overview based on current challenges and issues in the explainability of AI methods used in video classification in the medical field. The article is divided into four sections. In the first one, we summarize the rationale behind the field researched and intended audience. Then we summarize how we conducted the literature review. We introduce the explainability and interpretability of the AI aspect and the current requirements of explainability in the medical field including the metrics used for evaluation of XAI methods. The following section is dedicated to the particular XAI methods used to explain the decisions of the models in image and video processing tasks and explains selected XAI methods in more detail. The next section focuses more on the XAI methods used for deep learning video processing from different domains and suggests applying similar principles to video processing in healthcare.

## RATIONALE AND INTENDED AUDIENCE

In the medical environment, feature extraction from ultrasonography (USG), magnetic resonance imaging, computed tomography, X-ray, and other imaging modalities still heavily relies on radiologists' expertise. However, machine learning algorithms (ML) and deep learning models have been introduced over the past decades to aid this process; they often aid decision-making. Traditional ML approaches first extract hand-crafted features followed by application of classifiers such as support vector machine, decision trees, naive Bayes classifiers or K-nearest neighbours. However, these methods incorporate the shortcomings of hand-crafted features. They are not invariant to occlusion, illumination, morphological variation, rotation etc.

The interpretability and explainability of analytical models are becoming increasingly important, especially in the context of applications in the medical domain that strongly require credibility of deployed models. The problem becomes more complex when processing 2D image sequences or video sequences. The explainable techniques consider temporal and spatial information together and do not distinguish what role movement plays in decision-making with such data.

The article is intended to support academic and industry researchers working on deep learning in medical video analysis and the explainability of generated models. We expect our results to inspire the researchers to explore new methods improving explainability in close cooperation with relevant experts. Also, we expect practitioners to see the potential and benefits of deep learning models and will contribute their knowledge and experience to the final quality of models.

# SEARCH METHODOLOGY

The methodology used for the purpose of conducting this survey consisted of searching for information from general to more specific. We divided this procedure into four steps. In the first step, we focused on a general overview of the XAI area, its basic concepts, legislative requirements, and current trends in medicine. We mainly relied on articles providing an overview of XAI, which provided us with basic information about XAI and directed us to various aspects of XAI and legislative documents. In the second step, we focused on articles that use XAI methods in medicine. We looked for information on what requirements are essential for AI in medicine and its explanations. We focused mainly on research articles that used XAI methods to explain models in the field of medicine and health care. At the same time, we identified the problems related to the insufficient evaluation of the quality of XAI outputs. In the third step, we took a closer look at the metrics and possibilities for evaluating the quality of XAI methods and XAI methods that are used in processing image data in medicine in particular. Due to the scope of the article, we only describe selected XAI methods in the article. In the fourth step, we focused on the specific problem of using XAI methods in the processing of video data from the field of medicine.

In the entire search process (in all mentioned steps), we used Google Scholar to retrieve the relevant studies, as well as references from other survey articles. We used multiple queries consisting of keywords selected as relevant for a particular steps. The nature of our survey required to collect the articles not only from a specific area of XAI methods for deep learning-based video processing, but also related articles from medical imaging applications, as mentioned in the previous paragraph.

First, we collected the studies related to the basic aspects of XAI in the medical domain. We focused on collection of requirements and basic concepts applied in this domain. We used very common keywords ("XAI" and "healthcare" or "medical domain") to retrieve the documents. Then, in the second step, we collected the studies dedicated to particular XAI techniques applied in medical domain. Here we used a combination of keywords related to XAI, domain and methods. In a similar fashion we collected the studies describing the metrics related to XAI methods in medical domain. In the last step, studies about deep learning applied to video data in medicine were retrieved using following search procedure. The queries consisted of: (1) deep learning of any type (deep learning in general, CNN, LSTM or other architectures); (2) video data; (3) explainability or interpretability related keywords (or abbreviations).

The retrieved publications were screened by two reviewers, who performed relevance-based selection to select the studies considered to be eligible for the scope of this survey. During the selection process, we did not consider abstracts or in-progress reports; we removed duplicates (*e.g.*, articles from multiple sources). In all particular areas, we considered only studies from the medical domain, did not consider the methods and metrics not used in this domain and finally we focused on those dedicated to evaluation of one or multiple deep learning models for video processing in the medical domain and any aspect of explainability or interpretability. The investigated studies could demonstrate the available options for the video analytical tasks using deep learning methods in the medical
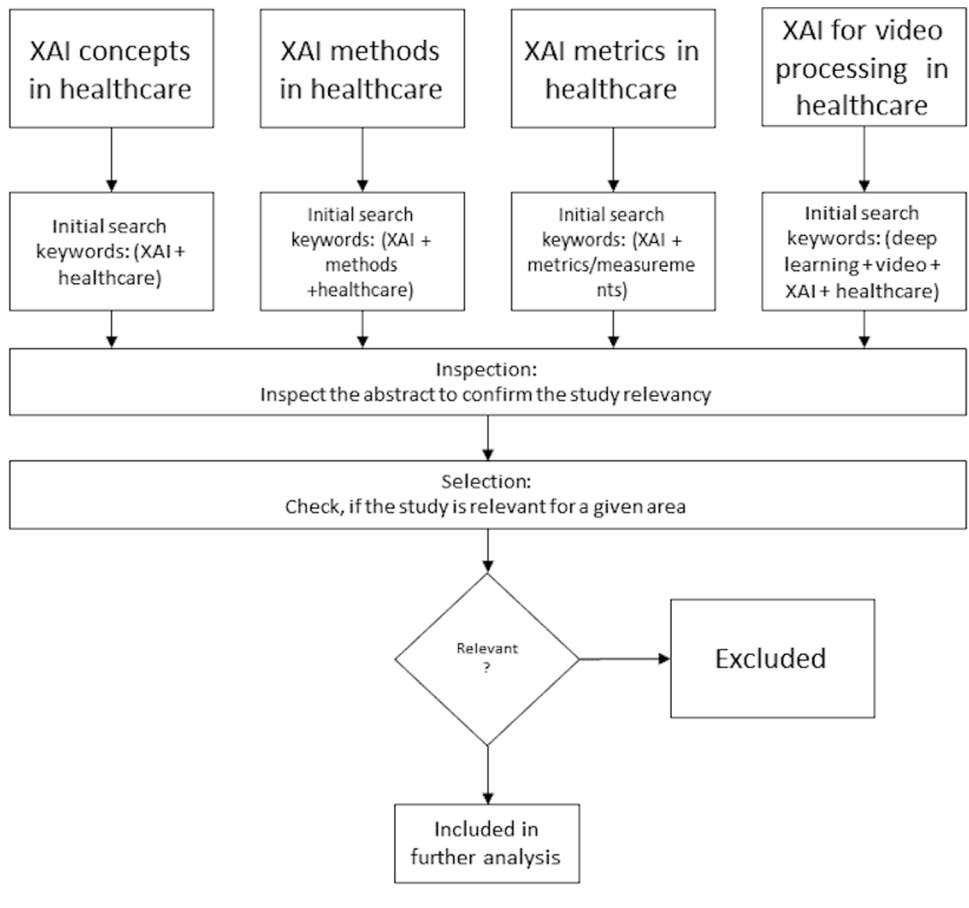

Figure 1  Survey methodology.

domain. The overall process of the survey methodology is depicted on Fig. 1. In total, this resulted in the retrieval of 87 articles, including books, articles, review articles, and journal and conference articles.

## EXPLAINABILITY AND INTERPRETABILITY

Interpretability and explainability are often used in the literature as synonyms, but many authors distinguish them. The term understanding is sometimes used as a synonym for interpretation and explanation in the context of XAI (*Das & Rad, 2020*). In this context, the term "understanding" usually means a functional understanding of the model instead of an algorithmic understanding of the model at a low level. Understanding tries to describe the outward behavior of a black-box model without trying to clarify its internal behavior.

In *Montavon, Samek & Müller (2018)* the authors distinguish between interpretation, which they define as the mapping of an abstract concept to a domain that can be perceived and understood by a human expert, and explanation, which they define as a set of interpretable features that contributed to the example of decision making. In *Edwards*

*& Veale (2017)*, the authors divided the explanations into model-centric, and object-centric, which basically correspond to the definitions of interpretability and explainability from *Montavon, Samek & Müller (2018)*. Similar tasks are explained in *Doshi-Velez & Kim (2017)* as global and local interpretability. These terms will be explained later on in the section XAI Methods. European Union (EU) legislation and the General Data Protection Regulations (GDPR), which deal with the processing of personal data, mention only the term explainability. Comprehensibility (*Lecue, 2020*) is used in the literature as a synonym for interpretability. In *Lipton (2018)* transparency is used as a synonym for the interpretability of the model, which is in a sense with understanding the logic of how the model works.

*Beaudouin et al. (2020)* explain the concept of explainability as "explain" with the suffix "-ability". Explainability becomes the ability to be explained. In the following chapters, we will therefore use the term explainability in this sense, covering alternatively interpretability (model-centric) and explainability (object-centric or local).

## Explainability as part of next-generation AI systems

The concept of explainability is increasingly found as one of the main requirements for AI systems in documentation. This may be as part of the requirements for the application domain, such as banking, healthcare, or they may be part of legislative regulations that are gradually coming along with the development of AI systems. The ethical aspect should be equally important, and they deal with the direct but also indirect impact of AI decisions on people's lives.

*Fjeld et al. (2020)* and *Healey (2020)* in their study analyzed 36 important documents about AI requirements from various fields, such as organizations or government documents or recommendations for AI, and based on these documents, defined eight key principles of contemporary AI, including the terms explainability and transparency under one of these principles:

- *Privacy*. AI systems should respect individuals' right to privacy, both in the use of data in technological systems and in the provision of data to decision-making agencies.
- *Accountability*. It is important that responsibility for the impacts of AI systems be properly defined and that remedial action is provided.
- *Safety and security*. AI systems must be secure and operate as designed. They also need to be secured and resilient against abuse by unauthorized parties.
- *Transparency and explainability*. AI systems should be designed and implemented to allow supervision as well as interpretation of activities in comprehensible output and to provide information on where, how, and when these systems are used. This principle is the response to challenges such as transparency, explainability, open source data and algorithms, or right to information.
- *Fairness and non-discrimination*. The principles of justice and non-discrimination require that AI systems should be designed and used to maximize fairness and minimize bias.
- *Human control of technology*. This principle requires that important decisions remain under human control all the time.

- *Professional responsibility*. This principle addresses the responsibilities and the role of individuals in the process of developing and deploying AI systems and calls for professionalism and integrity in ensuring communication with stakeholders on the long-term effects of these systems.
- *Promotion of human values*. The principles of human values state that the goals pursued by AI and how they are pursued should correspond with our values and generally support human well-being.

In addition to these key principles, which should become part of modern AI systems, many scientists, lawyers, and psychologists are currently dealing with ethical issues related to AI. Especially because with the increasing possibilities that AI offers us, new problems or questions arise, especially in applications that have a major impact on human lives. For example, how do we ensure that AI is fair and free from racial or gender prejudice? Who will be responsible if life is threatened due to an AI's decision? How to ensure that AI is fair and transparent? When can the AI decide by itself and when is it necessary to retain the supervision of a responsible person?

Recent initiatives in this area have also confirmed the importance of these problems. In the EU, the AI Expert Group has produced document the Ethics Guidelines for Trustworthy AI (*High-Level Independent Group on Artificial Intelligence (AI HLEG), 2019*), which provides guidelines for the development of trusted AI based on the principles of fundamental human rights that apply throughout the EU. The result is a kind of framework that defines four ethical principles:

1. *Respect for human autonomy*: A person has the right to supervise the system and to intervene in the AI process at any time.
2. *Prevention of harm*: This principle aims to prevent AI systems from harming a person, whether physically or mentally.
3. *Fairness*: The aim is to prevent discrimination or bias in AI.
4. *Explainability*: AI systems and their decisions should be explained in a way that is understandable to the stakeholders involved. Humans should know when they are using an AI system and must be informed about its capabilities and limitations.

Also, commercial companies engaged in research in AI applications are interested in creating systems that are ethical, fair, and transparent. For example, Google has released a document with its own principles that they want to follow when creating AI systems (*Pichai, 2018*).

China has similarly built on these ideas and, through the China Academy of Information and Communications Technology (CAICT), has issued a white paper on trustworthy AI (*China Academy of Information and Communications Technology JD Explore Academy, 2021*)—this is particularly noteworthy as it is in line with other major regulators in other countries.

From this point of view, transparency is an essential part of the creation and deployment of AI systems in the real environment and should be included in the design of the AI system. Of course, there are exceptions in this area as well, applications in which explainability does not play such an important role, especially business applications that focus on model

accuracy and the potential profit and for which time devoted to a deeper understanding of models would be cost-inefficient.

However, in safety-critical environments such as autonomous vehicles, industry, or healthcare, explainable methods are essential and required when deploying AI to help human decisions.

## XAI in healthcare

In the healthcare field, AI can be very beneficial. There are already practical deployments of AI, *e.g.*, to help doctors to identify the heart failure problems (*Choi et al., 2016*), lung problems after thoracic surgery (*Jaščur et al., 2021*) or automatic detection of COVID-19 from lung ultrasound (*Born et al., 2020*). However, the full potential of AI systems is limited by the inability of the majority of algorithms to explain their results and decisions to human experts. This is a huge problem, especially in the medical field, where doctors need to understand why AI has made a decision and how it came to that decision. Transparent algorithms could reasonably increase the confidence of medical experts in future AI systems (*Ahmad, Teredesai & Eckert, 2018*). Therefore, research aimed at creating XAI systems for medical applications requires the development of new methods for machine learning and human–computer interaction. There is a certain tension between the accuracy and explainability of machine learning methods. The most powerful models (especially deep learning (DL) or ensembles) are often least transparent, and methods that provide clear and comprehensible explanations known as interpretable models (*e.g.*, decision trees) are less accurate (*Bologna & Hayashi, 2017*).

In the healthcare domain, the motivation for using XAI methods is evident. In many cases, both end-users and the critical nature of the predictions require some transparency, either for user involvement or for patients' safety. XAI methods contribute significantly to transparency. However, sometimes an explanation of machine learning predictions is not enough. It is important to think about how the end-user interprets the results, how they are incorporated into the work process, or how they are used in other ways. Healthcare experts are often overwhelmed by the influx of patients, the influx of data about these patients, and the related tasks that are required of them, such as entering data into the system, analyzing available electronic health records, providing health care. Therefore, if AI systems and their explanations are not presented in the right way, it will not help healthcare experts, but on the contrary, it takes extra work. Hence, these systems should be created specifically tailored to the domain, and the perspective of the user who will work with them (*Ahmad, Teredesai & Eckert, 2018*).

AI is often associated with the idea that artificial intelligence should replace the decisions of health professionals. However, it is not obligatory to create systems in this way. Conversely, AI can be beneficial in important decisions that doctors must make, especially if the reasons for AI decisions or predictions are properly explained.

## Requirements of AI systems in the medical field

The field of medicine places specific requirements on all computer systems because it requires these systems to be safe, reliable, secure, certified, or audited. In addition, the

systems must work together and be fault-tolerant. A system error can cause a power outage or the administration of the wrong medication, resulting in the worst case in the death of a patient. It is, therefore, necessary that responsibility for the proper functioning of all systems is defined. This responsibility lies with the system administrators or certification authorities.

In the healthcare field, research focuses on the needs and specific requirements for security, trust, or accountability. AI's ethical or regulatory aspects in healthcare are also increasingly becoming a concern in this area. These concerns include, for example, model bias, lack of transparency, privacy concerns related to sensitive data used to train models, or liability issues. Although these concerns are often a topic of discussion, there are very few practical recommendations or examples.

A recent publication (*Reddy et al., 2020*) provides a governance model for AI in healthcare (GMAIH) that covers the introduction and implementation of AI models in health care. This model includes recent requirements from the United States Food and Drug Administration (FDA) (*Food and Drug Administration, 2016*) institute about requirements for AI systems. The GMAIH model outlines methods and practices for these four categories:

- Fairness—there should be data governance panels to oversee the collection and use of data. AI models should be designed to ensure procedural and distributive fairness.
- Transparency—includes transparency in decision-making on AI models and support for patient and physician autonomy.
- Credibility—education of physicians and patients in AI should be applied to enhance it. The integration of AI systems should include fully informed consent from patients to the use of AI and appropriate and authorized patient data.
- Accountability—means regulation and responsibility in the approval, implementation, and deployment phase of AI applications in healthcare.

Legislative requirements for AI systems in healthcare can vary from one part of the world to another. New AI systems and devices are subject to FDA approval in the US. In the EU, unlike the US, medical devices are not approved by a centralized agency. Medical devices are divided into risk classes (*Muehlematter, Daniore & Vokinger, 2021*), with the lowest risk class 1 being the device manufacturer's responsibility. Medical devices in the high-risk classes (IIa, IIb, and III) are dealt with by private 'notified bodies'—*i.e.,* organizations that have been accredited to carry out conformity assessment and issue the Conformité Européenne (CE) mark.

The FDA has only recently published (*US Food and Drug Administration (FDA), 2021*) the agency's first action plan for software as a medical device (SaMD) based on artificial intelligence/machine learning (AI/ML). This action plan describes a multi-pronged approach to advance the agency's oversight of AI/ML-based medical software. We can expect the EU will follow the US in improving oversight of AI/ML control of healthcare systems in the near future.

## Desiderata of XAI models

In the literature on explainability, we often come across the term "desiderata" which we could translate as necessary requirements for XAI methods. These requirements represent aspects or properties that are expected and required from a method capable of explaining AI models. These requirements also vary in the literature or are intended for specific types of methods, *e.g.*, Desiderata for gradient methods (*Das & Rad, 2020*) or Desiderata for interpretable model (*Guidotti et al., 2018*).

General requirements to be met by XAI models also include fidelity, or honesty (*Ribeiro, Singh & Guestrin, 2016*; *Plumb, Molitor & Talwalkar, 2018*). Other requirements include robustness or stability, which measures whether similar input instances generate similar conclusions (*Alvarez-Melis & Jaakkola, 2018*) as well as interpretability or comprehensibility (*Narayanan et al., 2018*), which means measures how difficult is for a person to understand the results from a given XAI model. Other requirements that were defined in *Robnik-Šikonja & Bohanec (2018)* for XAI methods are expressive power, translucency, portability, and algorithmic complexity. For individual explainability, authors defined other necessary properties such as accuracy, fidelity, consistency, comprehensibility, certainty, degree of importance, novelty, and representativeness.

However, these desiderata depend on the specific application or environment in which the models will be deployed. The authors of the article on the deployment of explainable models (*Bhatt et al., 2020*) argue that these requirements should be designed only based on the selected application and environment. It should be based on the following three points: 1. Identify stakeholders; 2. Involve each of the stakeholders; 3. Understand the reasons for an explanation.

A grand overview of desiderata based on different stakeholders was provided by the authors of the study (*Langer et al., 2021*). They divided the stakeholders into five classes: users, (system) developers, affected parties, deployers, and regulators. They created a list of 29 desiderata to which they assigned a stakeholder class and the articles where they appeared. This list is not definitive and will tend to change or expand over time.

Inspired by this overview, we collected and summarized recently published research articles and performed a similar overview for the medical field. The desiderata that appear in the field of medicine are summarized in the Table 1.

Based on the table we can say that the most frequent requirements for XAI methods in the medical field are accuracy, accountability, transparency and trust.

## XAI metrics and measurements

Based on requirements from the section on desiderata of XAI models, it is possible to compare models and select those that are suitable for the application we need, *e.g.*, in medicine (*Ahmad, Teredesai & Eckert, 2018*). However, recent practical approaches have shown that this comparison may not be sufficient and that more attention needs to be paid to practice tests along with evaluations from domain experts using these models (*Jesus et al., 2021*).

**Table 1  XAI desiderata in the medical field.**

| Desideratum | Description | Stakeholder | Occurence |
|---|---|---|---|
| acceptance | Improve acceptance of systems | Deployer, Regulator | *Reddy et al. (2020)* and *Panigutti, Perotti & Pedreschi (2020)* |
| accountability | Provide appropriate means to determine who is accountable | Regulator | *Reddy et al. (2020)*, *Panigutti, Perotti & Pedreschi (2020)*, *US Food and Drug Administration (FDA) (2021)*, *Ahmad, Teredesai & Eckert (2018)*, *Dave et al. (2020)* and *Tjoa & Guan (2019)* |
| | | | *Pawar et al. (2020b)* and *Larasati & DeLiddo (2020)* |
| accuracy | Assess and increase a system's predictive accuracy | Developer | *Reddy et al. (2020)*, *Ahmad, Teredesai & Eckert (2018)*, *Dave et al. (2020)*, *Tjoa & Guan (2019)*, *Khedkar et al. (2019)* and *Holzinger et al. (2017)* |
| | | | *Singh, Sengupta & Lakshminarayanan (2020)*, *Pawar et al. (2020a)*, *Brunese et al. (2020)*, *Alshazly et al. (2021)* and *Wei et al. (2020)* |
| autonomy | Enable humans to retain their autonomy when interacting with a system | User | *Reddy et al. (2020)*, *Holzinger et al. (2017)* and *Singh, Sengupta & Lakshminarayanan (2020)* |
| confidence | Make humans confident when using a system | User | *Reddy et al. (2020)*, *Larasati & DeLiddo (2020)*, *Holzinger et al. (2017)* and *Singh, Sengupta & Lakshminarayanan (2020)* |
| controllability | Retain (complete) human control concerning a system | User | – |
| debugability | Identify and fix errors and bugs | Developer | *Ahmad, Teredesai & Eckert (2018)*, *Dave et al. (2020)*, *Khedkar et al. (2019)*, *Holzinger et al. (2017)* and *Brunese et al. (2020)* |
| education | Learn how to use a system and system's peculiarities | User | *Reddy et al. (2020)* |
| effectiveness | Assess and increase a system's effectiveness; work effectively with a system | Developer, User | *Reddy et al. (2020)*, *US Food and Drug Administration (FDA) (2021)*, *Holzinger et al. (2017)*, *Brunese et al. (2020)* and *Alshazly et al. (2021)* |
| fairness | Assess and increase a system's (actual) fairness | Affected, Regulator | *Reddy et al. (2020)*, *Panigutti, Perotti & Pedreschi (2020)*, *Ahmad, Teredesai & Eckert (2018)*, *Dave et al. (2020)* and *Holzinger et al. (2017)* |
| informed consent | Enable humans to give their informed consent concerning a system's decisions | Affected, Regulator | *Reddy et al. (2020)* and *Wei et al. (2020)* |
| legal compliance | Assess and increase the legal compliance of a system | Deployer | – |
| ethics | Assess and increase a system's compliance with moral and ethical standards | Affected, Regulator | *Reddy et al. (2020)*, *Holzinger et al. (2017)*, *Tjoa & Guan (2019)* and *Singh, Sengupta & Lakshminarayanan (2020)* |
| performance | Assess and increase the performance of a system | Developer | *Reddy et al. (2020)*, *Panigutti, Perotti & Pedreschi (2020)*, *US Food and Drug Administration (FDA) (2021)*, *Ahmad, Teredesai & Eckert (2018)*, *Dave et al. (2020)* and *Khedkar et al. (2019)* |
| | | | *Singh, Sengupta & Lakshminarayanan (2020)*, *Pawar et al. (2020a)* and *Brunese et al. (2020)* |

**Table 1** (*continued*)

| Desideratum | Description | Stakeholder | Occurence |
|---|---|---|---|
| privacy | Assess and increase a system's privacy practices | User | *Reddy et al. (2020), Ahmad, Teredesai & Eckert (2018), Holzinger et al. (2017), Amann et al. (2020)* and *Larasati & DeLiddo (2020)* |
| responsibility | Provide appropriate means to let humans remain<br><br>responsible or to increase perceived responsibility | Regulator | *Reddy et al. (2020), Tjoa & Guan (2019)* and *Muehlematter, Daniore & Vokinger (2021)* |
| robustness | Assess and increase a system's robustness<br><br><br><br><br><br>(e.g., against adversarial manipulation) | Developer | *Reddy et al. (2020), US Food and Drug Administration (FDA) (2021), Tjoa & Guan (2019), Singh, Sengupta & Lakshminarayanan (2020), Alshazly et al. (2021)* and *Wei et al. (2020)*<br><br>*Muehlematter, Daniore & Vokinger (2021)* and *Muddamsetty, Jahromi & Moeslund (2021)* |
| security | Assess and increase a system's security | All | *Ahmad, Teredesai & Eckert (2018), Larasati & DeLiddo (2020), Holzinger et al. (2017), Brunese et al. (2020)* and *Amann et al. (2020)* |
| safety | Assess and increase a system's safety | Deployer, User | *Reddy et al. (2020), Ahmad, Teredesai & Eckert (2018), Holzinger et al. (2017), Singh, Sengupta & Lakshminarayanan (2020), Muehlematter, Daniore & Vokinger (2021)* and *Born et al. (2020)* |
| satisfaction | Have satisfying systems | User | – |
| science | Gain scientific insights from the system | User | *US Food and Drug Administration (FDA) (2021), Tjoa & Guan (2019)* and *Muehlematter, Daniore & Vokinger (2021)* |
| transferability | Make a system's learned model transferable to other contexts | Developer | *Alshazly et al. (2021)* |
| transparency | Have transparent systems | Regulator | *Reddy et al. (2020), Panigutti, Perotti & Pedreschi (2020), US Food and Drug Administration (FDA) (2021), Ahmad, Teredesai & Eckert (2018), Dave et al. (2020)* and *Tjoa & Guan (2019)*<br><br>*Pawar et al. (2020b), Larasati & DeLiddo (2020), Holzinger et al. (2017), Amann et al. (2020), Muehlematter, Daniore & Vokinger (2021)* and *Muddamsetty, Jahromi & Moeslund (2021)* |
| trust | Have appropriate trust in the system | User, Deployer | *Reddy et al. (2020), Panigutti, Perotti & Pedreschi (2020), US Food and Drug Administration (FDA) (2021), Ahmad, Teredesai & Eckert (2018), Dave et al. (2020)* and *Pawar et al. (2020b)*<br><br>*Khedkar et al. (2019), Larasati & DeLiddo (2020), Holzinger et al. (2017), Singh, Sengupta & Lakshminarayanan (2020)* and *Pawar et al. (2020a)* |
| trustworthiness | Assess and increase the system's trustworthiness | Regulator | *Reddy et al. (2020)* and *Dave et al. (2020)* |
| usability | Have usable systems | User | *US Food and Drug Administration (FDA) (2021), Tjoa & Guan (2019), Holzinger et al. (2017)* and *Amann et al. (2020)* |
| usefullness | Have useful systems | User | *Alshazly et al. (2021)* |
| verification | Be able to evaluate whether the system does<br><br>what it is supposed to do | Developer | *Tjoa & Guan (2019), Brunese et al. (2020)* and *Amann et al. (2020)* |

It is also possible to compare explainable methods from the point of view of several levels. The authors in *Doshi-Velez & Kim (2017)* propose three main levels for the evaluation of interpretability:

- Application level evaluation (real task): Implementation of models for explainability in a specific application and testing it on a real task. For example, software that will detect fracture sites based on X-ray records. The doctor could evaluate the quality of the explanations that the software offers to explain its intentions.
- Human-level evaluation (simple task): This level of explainability is also within applications, but the evaluation quality is not performed by experts, but by ordinary people - testers who are cheaper and also choose explanations according to how they help them understand at their level of knowledge.
- Function level evaluation (proxy task): This level does not require people. It is appropriate if a class of methods that the target class can work with is used, *e.g.*, a decision tree. This model can be bounded to better explainability, *e.g.*, using the decision tree pruning method.

However, the way in which methods are evaluated can vary considerably, depending on different objectives of their deployment, the stakeholders for which they are intended, and the type of the method used. This was also noted by *Mohseni, Zarei & Ragan (2018)* who categorized metrics based on design goals and evaluation metrics. They categorized requirements by type of target user into the following three groups:

- AI novices—users with little expertise on AI models but using AI systems daily. XAI goals for this group of users are: *algorithmic transparency, user trust and reliance, bias mitigation, privacy awareness*
- Data experts—data scientists or domain experts who use machine learning models for analysis and decision making tasks. Their goals are: *model visualization and inspection, model tuning and selection*
- AI experts—machine learning scientist, designers and developers of ML algorithms with their goals: *model interpretability* and *model debugging*

The model measurements can be divided as follows:

1. Computational measures

   - Fidelity of interpretability method (AI experts)—uses two metrics (*Velmurugan et al., 2021*), recall ($R = \frac{|TF \cap EF|}{|TF|}$) and precision ($P = \frac{|TF \cap EF|}{|EF|}$), where the term True Features (TF) represents the relevant features as extracted directly from the model and Explanation Features (EF) represents the features characterised as most relevant
   - Model trustworthiness (AI experts)—represents a set of domain specific goals such as safety (by robust feature learning), reliability, and fairness(by fair feature learning). Different similarity metrics, such as Intersection over Union (IoU) and mean Average Precision (mAP), are used to quantify the quality of model saliency explanations or bounding boxes compared to the ground truth (*Mohseni, Zarei & Ragan, 2018*). These metrics often depend on the model used and are compared to the annotated explanations.

2. Human-grounded measures

- Human-machine task performance (data experts and AI novices)—XAI should assist users in tasks involving machine learning. Therefore, it is important to measure user performance when evaluating XAI methods. For example, we can measure users' performance in terms of success rates and task completion times while evaluating the impact of different types of explanations.
- User mental model (AI novices)—The mental model represents how users understand the system. XAI assists users in creating a mental model of how AI works. One way of exploring these models is to ask users directly about their understanding of the decision-making process. The mental model can be measured by several metrics, *e.g.*, ease of users' self-explanations, user prediction of model output, or user prediction of model failure.
- User trust and reliance (AI novices)—User trust and reliability can be measured by explicitly gauging users' opinions during and after working with the system, which can be through interviews and questionnaires.
- Explanation usefulness and satisfaction (AI novices)—The effort is to identify user satisfaction and the usefulness of machine explanation. Various subjective and objective measures of understandability and usefulness are used to assess the value of the explanation to users. Qualitative evaluations in the form of questionnaires and interviews are most commonly used.

However, there is a lack of use cases for evaluating XAI methods in healthcare. In some articles (*Lauritsen et al., 2020*) the evaluation was carried out by manual inspection with domain experts. There are some articles (*Muddamsetty, Jahromi & Moeslund, 2021*; *Panigutti, Perotti & Pedreschi, 2020*) where the authors tried to evaluate and compare used XAI methods using computational measures.

In *Panigutti, Perotti & Pedreschi (2020)*, the authors developed a new model of explainability of black box models for processing sequential, multi-label medical data. To evaluate it, they used the computational measure's fidelity to the black-box, hit (tells if the interpretable classifier predicts the same label as the black-box), and explanation complexity while comparing the black-box model with its interpretable replacement in the form of decision rules.

However, the selection of appropriate metrics depends not only on the target domain or the method used but also on the type of data processed. In *Muddamsetty, Jahromi & Moeslund (2021)*, the authors investigated expert-level evaluation of XAI methods in the medical domain on an image dataset. In doing so, they used the state-of-the-art metrics AUC-ROC curve and Kullback–Leibler divergence (KL-DIV), comparing the results of eye-tracking expert observations against the results of XAI methods. They showed that it is important to use domain experts when evaluating XAI methods, especially in a domain such as medicine.

In a recent study (*Gunraj, Wang & Wong, 2020*), a new method called GSInquire was used to create heatmaps from the proposed COVID-net model for detection of COVID-19 from chest X-ray images. Together with the new method, the authors proposed new

metrics—impact coverage and impact score. Impact coverage was defined as coverage of adversarially impacted factors in the input. The impact score was defined as a percentage of features that impacted the model's confidence or decision.

# XAI METHODS

Due to the growing number of methods in the field of explainability, it is difficult to understand the advantages, disadvantages, or competitive advantages in different domains. There are different taxonomies of XAI methods (*Gilpin et al., 2018*; *Barredo Arrieta et al., 2020*; *Molnar, 2018*), but most of them agree on classifying methods into categories such as global methods (which explain the behavior of the model on the whole data set), local methods (which explain the prediction or decision for a specific example), ante-hoc (where the explanation model is created in the AI training phase), post-hoc (where the explanation model is created only on trained models), surrogate (an interpretable model replaces the AI model) or a directly interpretable model (decision trees or decision rules) is used. Molnar, in his book (*Molnar, 2018*) generally categorizes XAI methods into three types: (1) methods with internal interpretation, (2) model agnostic methods, and (3) example-based explanation methods. Another taxonomy of XAI methods is based on the data type (*Bodria et al., 2021*), such as tabular data, image data, and text data. Figure 2 depicts a commonly used categorization of the XAI methods *Linardatos, Papastefanopoulos & Kotsiantis (2021)*.

In this article, only selected methods used in video processing tasks will be explained and referred to in the text.

## Model agnostic methods

Model agnostic methods separate the explanations from the machine learning model. This brings an advantage over model-specific methods in their flexibility (*Ribeiro, Singh & Guestrin, 2016*) and universality. Agnostic methods can be used for a wide range of machine learning models, such as ensemble methods or deep neural networks. Even the output of an XAI method, whether it is a visual or textual user interface, also becomes independent of the machine learning model used. A single agnostic method can explain each of the multiple trained machine learning models and help decide the most appropriate deployment model. These methods can be further divided into global and local methods. Global methods describe the impact of features on the model on average, and local methods explain the model based on the predictions of individual examples.

### SHAP

SHAP (SHapley Additive exPlanations) by *Lundberg & Lee (2017)* is a method for explaining individual predictions of the model. This method is based on Shapley values the game theory.

*Shapley (2016)* invented Shapley values as a way of providing a fair solution to the following question: If we have a coalition $c$ that collaborates to produce value $v$, how much did each individual member contribute to the final value?

To find the answer to this question, we can compute a Shapley value for each member of the coalition. For example, if we want to find the Shapley value for the first member.

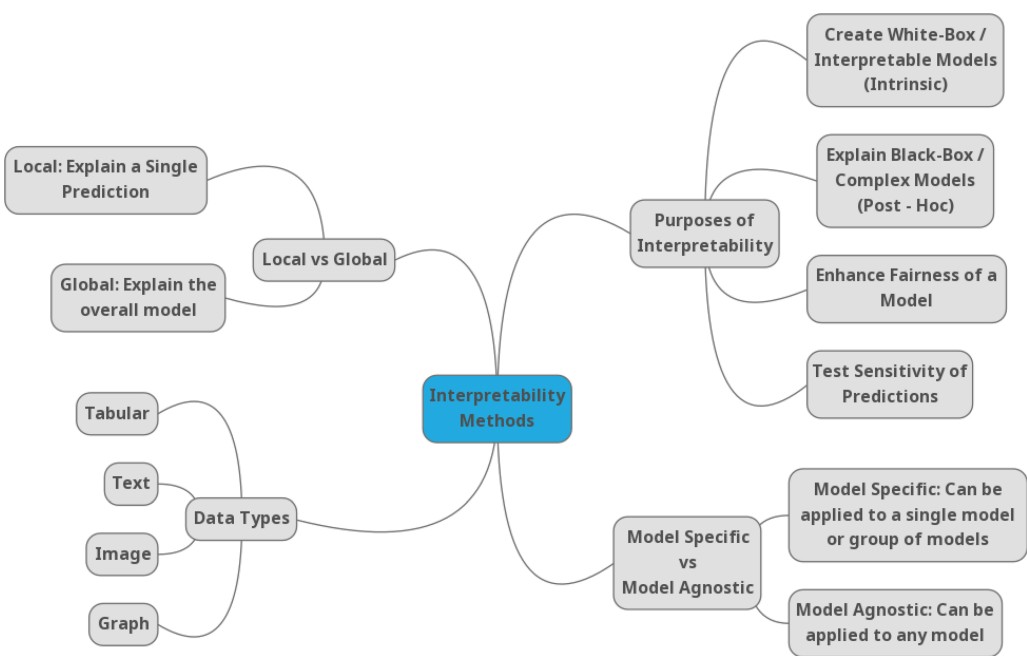

**Figure 2** Taxonomy of the XAI methods according to *Linardatos, Papastefanopoulos & Kotsiantis (2021).*

Let us compare a coalition formed with all members and a coalition formed without the first member. The difference between these results is the marginal contribution of the first member for the coalition composed of the other members. We then look at all the marginal contributions we get in this way. The Shapley value is the average of these results for a single member. We can repeat this process for all members (*Shapley, 2016*).

SHAP is based on a similar idea. Unlike coalition members, it looks at how individual features contribute to a model's outputs. However, it does this in a specific way. As the name implies, the method uses Shapley values for explanations, but it also uses additive features. *Lundberg & Lee (2017)* define an additive feature attribution as follows: If we have a set of inputs $x$ and model $f(x)$, we can define a set of simplified local inputs $x'$ and we can also define an explanatory model $g(x')$.

What we need to ensure is:

1. if $x'$ is roughly equal to $x$, then $g(x')$ should be roughly equal to $f(x)$,
2. $g(x') = \phi_0 + \sum_{i=1}^{N} \phi_i x'_i$

where $\phi_0$ is the average output of the model and $\phi_i$ is the explained effect of feature $i$, how much feature $i$ changes the model, and this is called it's attribution. In this way, we can get a simple interpretation for all features.

SHAP describes the following three desirable properties:

1. Local accuracy—if the input and the simplified input are roughly the same, then the actual model and the explanatory model should produce roughly the same output.
2. Missingness—if the feature is excluded from the model, it's attribution must be zero.

3.  Consistency—if the feature's contribution changes, the feature effect cannot change in the opposite direction.

SHAP satisfies all three properties. The problem occurs, when computing Shapley values. There must be calculated values for each possible feature permutation. This means we need to evaluate the model multiple times. The get around this problem *Lundberg & Lee (2017)* devise the Shapley kernel or KernelSHAP.

KernelSHAP approximates Shapley values through much fewer samples. There are also other forms of SHAP presented in *Lundberg & Lee (2017)*: Low-Order SHAP, Linear SHAP, Deep SHAP, Max SHAP. However, KernelSHAP is the most universal and can be used for any type of ML model.

For visualization of SHAPley values, we can use a summary plot. Each point of the graph on the $x$-axis represents a Shapley value for one element of *Molnar (2018)*. The elements on the $y$-axis are sorted by importance. The color represents the feature value from low (blue) to high (red). For example, from the Fig. 3, a low number of years of contraceptive use reduces the risk of cancer. Conversely, a high number of years increases this risk.

### LIME

In their work, *Ribeiro, Singh & Guestrin (2016)* proposed a method called Local Interpretable Model-agnostic Explanations (LIME). As the name implies, it is a method that focuses on local interpretation and is universal concerning the model used. LIME is a method that uses a surrogate for the black-box model in the form of an interpretable model, which it constructs based on examples within the neighborhood of the observed example and approximates the black-box model's predictions. This assumes that a simple interpretable model can explain the model's behavior in its neighborhood.

This principle is quite intuitive. We have a black-box model whose decisions we want to understand. We choose a single example and start creating variations of the features of the chosen example that we give to the model. We save the input data (variations) and the predictions of the black-box model. LIME will then train an interpretable model based on this data. This model should have a good approximation of the predictions, close to the black-box model, but this does not mean that it will also be a global approximation of the model. Therefore, this is one of the local models. Any interpretable model from the previous chapter can be used as an interpretable model.

In his book (*Molnar, 2018*), Molnar describes the process of the LIME method in steps:

- Choosing an example to explain black-box prediction.
- Creation of variations of the input data from the desired example.
- Allocation of weights by a new example. The example that is more similar to the desired example gets more weight.
- Training the chosen interpretable model on new variations of the weighted input data.
- Explanation of prediction using the trained interpretable model.

The LIME method can be applied to different types of input data, such as tabular data, text data, or images. The principle is the same, but the output differs in the interpretation of the outputs.

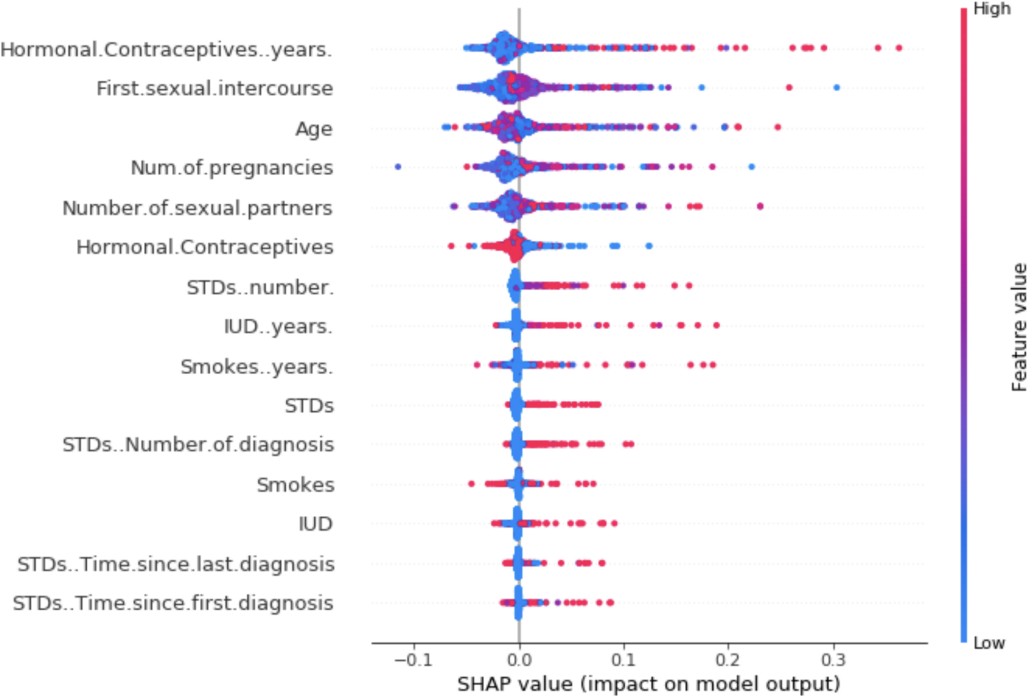

**Figure 3** SHAP summary plot (*Molnar, 2018*).

### TCIU

The Contextual Importance and Utility (CIU) (*Anjomshoae, Främling & Najjar, 2019*; *Anjomshoae, Kampik & Främling, 2020*) method explains the model's outcome using two algorithms Contextual Importance (CI) and Contextual Utility (CU). CI approximates the overall importance of a feature in the current context. CU provides an estimation of how favorable or not the current feature value is for the given output class. This can help to justify why one class is preferred over another. Explanations have contextual capabilities, which means that one feature can be more important for a decision about one class but irrelevant for another class. CI and CU values are formally defined as:

$$CI = \frac{Cmax_x(C_i) - Cmin_x(C_i)}{absmax - absmin}$$

$$CI = \frac{y_{i,j} - Cmin_x(C_i)}{Cmax_x(C_i) - Cmin_x(C_i)}$$

- $x$ is the input(s) (vector) for which CI and CU are calculated,
- $Cmax$ and $Cmin$ are highest and the lowest output values observed by varying the value of the input(s) $x$,
- $absmax$ and $absmin$ specify the value range for the output $j$ being studied.
- $y_{i,j}$ is the otuput value for the output $j$ studied when the inputs are those defined by $C_i$

## Model-specific explanations

There are several XAI methods in this group working with specific DL models, *e.g.*, CNN, LSTM, or GAN for image processing (*Alshazly et al., 2021*) or video processing models (*Chittajallu et al., 2019*). There are also XAI methods for specific data types like text, voice, or timeseries.

Papastratis, in his recent survey (*Papastratis, 2021*) presents current trends in explainable methods for deep neural networks. Some of the methods he presents have already been described above and belong to one of the previous categories. Papastratis has divided these methods into three categories:

- Visual XAI methods: visual explanations and plots
- Mathematical or numerical explanations
- Textual explanations, given in text form

### *Class Activation Mapping (CAM)*

CAM (*Zhou et al., 2016*) represents one of the basic methods from the visual domain. Other methods are also based on its principle. CAM adds a global average pooling layer between the last convolutional layer and the final fully connected layer of the CNN neural network. The fully connected layer, controlled by the softmax activation function, subsequently provides us with the desired probabilities at the output. We can obtain the importance of the weights concerning the category by back projecting the weights onto the saliency maps of the last convolutional layer. That allows us to visualize the CNN features from the layer responsible for the classification.

A mathematical formulation of CAM: Let $f(x,y)$ be the activation map of unit $u$ in the last convolutional layer at spatial location $(x,y)$. The result of the global average pooling (GAP) layer (injected between the last convolutional layer and the final fully connected layer) is represented as:

$$F_u = \sum_{x,y} f_u(x,y).$$

For a class $c$, an input to softmax will be:

$$S_c = \sum_u w_u^c F(u).$$

Output of softmax layer:

$$P_c = \frac{e^{S_c}}{\sum_c e^{S_c}}.$$

Thus, the final equation for an activation map of class c would be:

$$M_c(x,y) = \sum_u w_u^c f_u(x,y).$$

CAMs are a good and simple technique for interpreting features from CNN models. The disadvantage of this method is noise which causes a loss of spatial information. CAMs require a CNN model that contains a GAP layer, and CAM heatmaps can be generated only for the last convolutional layer. Therefore, other algorithms such as Grad-CAM have been developed.

### Gradient-weighted Class Activation Mapping (Grad-CAM)

Grad-CAM (_Selvaraju et al., 2019_) is a generalization of CAM, which can be applied to any type of CNN. Grad-CAM is applicable to different types of CNN architectures: CNN, VGG, DenseNet. Grad-CAM does not require a GAP layer and can be used for heatmaps for any layer. The difference between CAM and Grad-CAM is in calculating the weights for each heatmap. Grad-CAM takes the convolutional layer's feature map and calculates which attribute is important, based on the gradient of the score, at the selected target class. Then the neuron weights are obtained by global averaging of the gradients. In this way, we obtain the weights of the flags for the target class. By multiplying the feature maps with their weights we obtain a heatmap highlighting regions that positively or negatively affect the class of interest. Finally, we apply the ReLU function, which sets the negative values to 0 because we are only interested in the positive contributions of the selected class. In this way, we obtain feature maps that highlight important regions of the input image for the selected target class.

Visualization methods like Grad-CAM can help identify bias in the trained model, as shown in Fig. 4. The activation maps showed part of the image that the model uses. The model decisions are based on the edge of the image instead of the lung area.

Table 2 summarizes the advantages and disadvantages of the described methods.

## XAI IN VIDEO PROCESSING APPLICATIONS

Deep learning methods perform very well in image processing and visualization tasks. With the increasing performance of AI computing units and decreasing cost, deep learning methods are also becoming more applicable in video processing, which is computationally more complex. However, video can provide important information about the evolution of the area under study over time. Thus, we can track the movement of objects or the temporal appearance of an object, which cannot be obtained simply from images. As the complexity of the neural network for video processing increases, the problem of the explainability of these networks also increases.

XAI methods for video processing applications are based on visualization methods for 2D image processing. The most common methods are CAM and Grad-CAM, which are adapted for 3D neural networks, or methods that combine visual information with textual information.

In the following subsections, we will discuss current approaches for using XAI methods for DL video processing, and potential applications of XAI methods in medical video processing.

### XAI for deep learning video processing

_Hiley et al. (2020)_ in their work, focused on explaining the relevance of motion for activity recognition. They point out that in the same way, there are attempts to adapt XAI methods initially developed for images to enable them to work with videos (3D inputs). Similarly, 3D convolutional neural networks are being adapted to work with video in this way. However, the methods adapted in this manner consider spatial and temporal information together. Therefore, when using these XAI methods, it is impossible to clearly distinguish

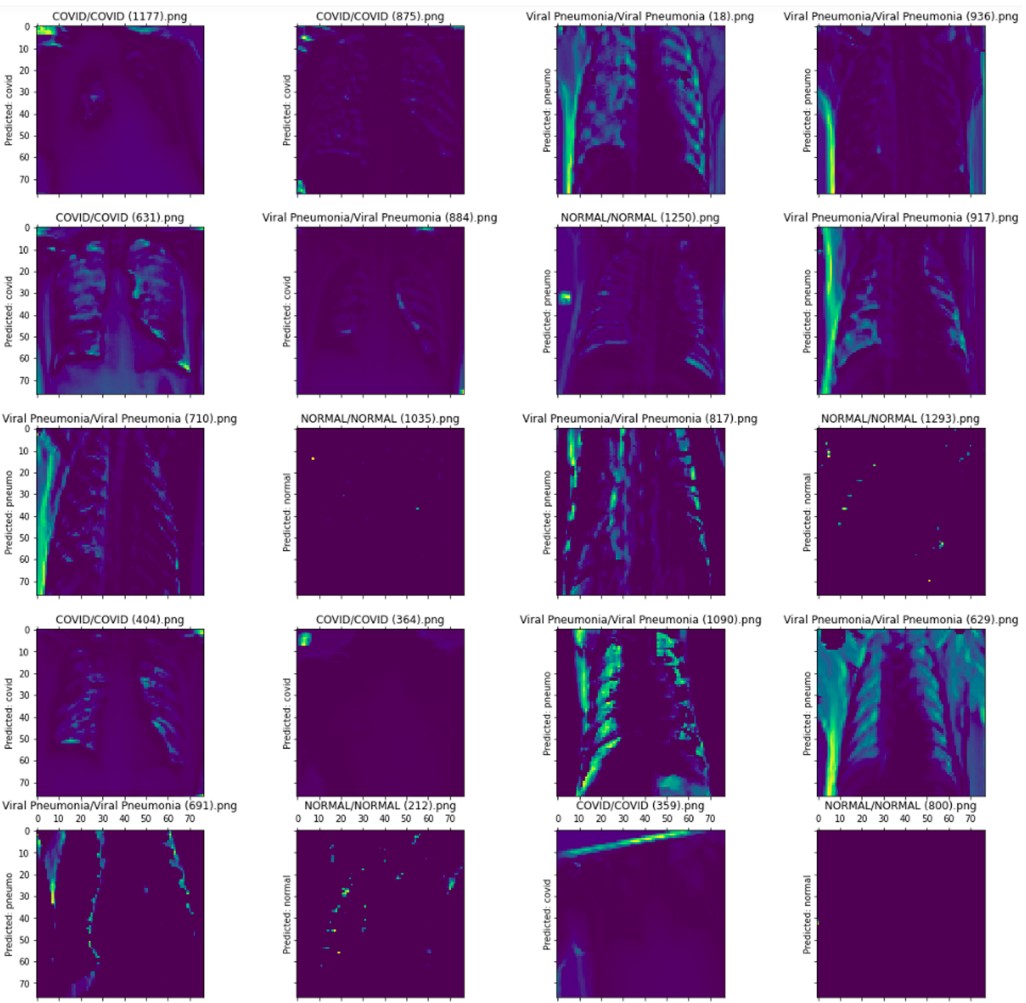

**Figure 4** **Differences in Grad-CAM visualization between a biased and unbiased model (*Moreau, 2018*).**

the role of motion in 3D model decision-making. The problem is that these methods do not consider motion information over time. Therefore, the authors proposed a method, selective relevance, for adapting 2D XAI methods for motion tracking and these are are better understood by the user. They demonstrated the results using several XAI methods and observed the improvement of the explanation for motion over time. Their method offers a different perspective to explain the model decision-making in video classification and it improves the explanations offered. A comparison of 3D and selective methods can be seen in Fig. 5. From the left, there are: original video frame, 3D DTD (Deep Taylor decomposition), selective DTD, 3D Grad-CAM, selective Grad-CAM, 3D guided backprop explanation, 3D guided GradCAM explanation, selective guided GradCAM. In selective DTD, the resulting explanations are more focused and simpler compared to 3D DTD. In selective Grad-CAM, the center of focus remains stable but the edges are stronger with

**Table 2  Comparison of XAI methods.**

| Method | Advantages | Disadvantages |
|---|---|---|
| SHAP | SHAP predictions are fairly distributed among symptom values. It is an agnostic method regarding the model used. Fast computation when applied to tree-based models. Allows both local and global interpretation of the model. | KernelSHAP is slow. For global interpretation, this requires computing many instances. Counting for many values is, therefore, slow and impractical. SHAP values can be misinterpreted. It is possible to create deliberately misleading interpretations. As an end user, you, therefore, cannot be 100% sure of the veracity. |
| LIME | It can be applied directly to a trained interpretable model (any trained model). It can be applied to tabular, textual, or image data. It uses a metric for the goodness-of-fit measure that also tells how well the model approximates the black-box model around the example we are interested in. | When used with tabular data, defining what a neighborhood means is difficult. The complexity of the model is defined in advance. The user chooses between fidelity and sparsity of explanation. The stability of explanations - with two close examples; LIME may offer different explanations. |
| CIU | CIU enables explanation of why a certain instance is preferable to another one, or why one class (outcome) is more probable than another. CI and CU values can be calculated for more than one input which means that higher-level concepts can be used in explanations. It is also a lightweight method which makes the model run faster compared to LIME and SHAP. | CIU is a novel approach and still in an early stage of development compared to LIME or SHAP. |
| CAM | CAMs are a good and simple technique for interpreting features from CNN models. CAM does not require a backward pass through the network again. | The noise causing a loss of spatial information. CAM heatmaps can be generated only for the last convolutional layer. It cannot be used for computer vision tasks such as visual question answering. |
| Grad-CAM | Based on the gradients of the task-specific output with respect to the feature maps, Grad-CAM can be used for all computer vision tasks such as visual question answering and image captioning. It uses the gradients of the output score as the weights of the feature maps that eliminates the need to retrain the models. | When there are multiple occurrences of the target class within a single image, the spatial footprint of each of the occurrences is substantially lower. The inability to localize multiple occurrences of an object in an image. Inaccurate localization of heatmap with reference to coverage of class region due to the partial derivatives premise. |

red areas representing higher intensity change. The last three methods do not provide comparable results.

*Nourani et al. (2020)* presented research focusing on perceptions of AI systems influenced by first experiences and how explainability can help users to form an idea of the system's capabilities. They used a custom neural network to recognize activities from video and looked at whether the presence of explanatory information for system decisions influences the user's perception of the system. They tested how changing the order of the model's weaknesses and strengths can affect users' mental models. They found that the first impression of the system can significantly impact the task error rate and the user's perception of the accuracy of the model. Adding additional explanations was not enough to negate the influence of first impressions, and users with a negative first impression also tried to find errors in further explanations. In contrast, users with positive first impressions were more likely to ignore errors in explanations.

*Escalante et al. (2017)*, created a challenge for using DL and XAI methods for automated recruitment of people based on their videos. When interviewing, it is often the case that selection is based on subjective feelings and first impressions rather than objective

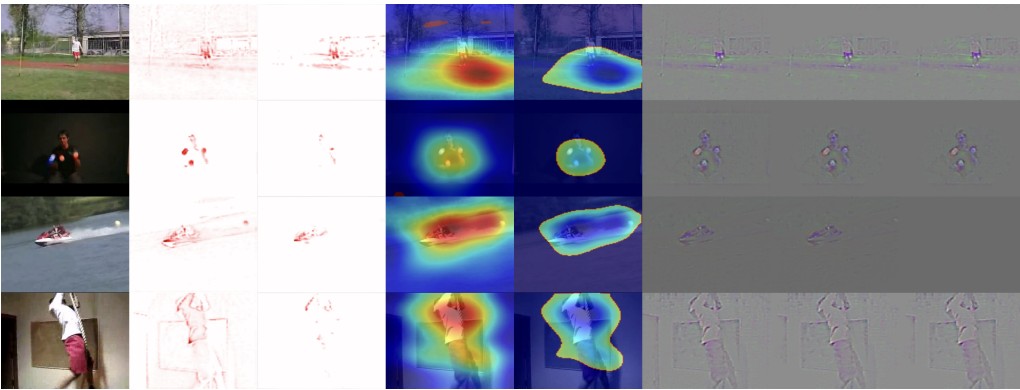

**Figure 5 Comparison of XAI methods for activity recognition from video dataset UCF-101.** From top to bottom: classes JavelinThrow, JugglingBalls, Skijet and RopeClimbing. From Left to Right: Original frame for reference, 3D DTD explanation, selective DTD, 3D GradCAM explanation, selective GradCAM, 3D guided backprop explanation, 3D guided GradCAM explanation, selective guided GradCAM (*Hiley et al., 2020*).

assessment, which can lead to bias. In their study, they highlight the problem of explaining models' decisions and using XAI to identify important visual aspects, trying to understand how these aspects relate to the model's decisions, and gaining insight into unwanted biases. Their goal is to increase the awareness and importance of XAI methods for machine decision-making applications such as recruitment automation. The study describes the environment, scenario, and evaluation metrics. These are short videos (15s) of job recruitment interviews. This challenge resulted in several different models in XAI methods.

In their work, *Stano, Benesova & Martak (2020)* presented a novel approach for explaining and interpreting the decision-making process to a human expert working with a convolutional neural network-based system. In their work, they used Gaussian mixture models (GMMs) for a binary code in vector space that describes the process of input processing by a CNN network. By measuring the distance between pairs of examples in this perceptual encoding space, they obtained a set of perceptually most similar and least similar samples, which helped clarify the CNN model's decision.

This approach can be applied to 3D objects such as magnetic resonance imaging (MRI) or computed tomography (CT). 3D objects are very similar to videos; however, their third dimension is constant, unlike videos whose third dimension can be variable. The proposed method is suitable for explaining the model to medical personnel through similar examples from the same domain.

## Deep learning video processing in medical applications

*Ouyang et al. (2020)* from Stanford University created a new DL model based on echocardiography videos, which they called EchoNet-Dynamic. Repeated human measurements confirm that the model has a variance smaller than that of human experts, who need years of practice to make a correct assessment and it outperforms human experts in the tasks of cardiac left ventricle segmentation, ejection fraction estimation, and cardiomyopathy assessment. The model can quickly identify changes in ejection

fraction and can be used as a basis for the real-time prediction of cardiovascular disease. Along with the article, they also published more than 10,000 annotated echocardiographic videos. *Born et al. (2020)* aimed to help physicians diagnose COVID-19 using AI. In doing so, they used an image from a lung ultrasound. Ultrasound is non-invasive and commonly present in medical facilities around the world. Their contribution can be described in three points. They collected a set of ultrasound data compiled from various online sources and published it publicly. The dataset contains 64 videos from which 1,103 images were created, divided into three classes (654 COVID-19, 277 pneumonia, 172 healthy controls). Second, they created a DL model of the POCOVID-Net convolutional network that achieves an accuracy of 89%. Third, they provided a web service (https://github.com/jannisborn/covid19_ultrasound/tree/master/pocovidscreen) on which the POCOVID-Net model is deployed and it enables physicians to make predictions based on ultrasound images or upload their own images to contribute to the dataset extension. This work would be even better if the system could also provide explanations for its decisions. XAI methods would increase physician confidence and make the system more transparent. In the current pandemic situation, this system has great potential to help identify COVID-19.

In some cases, lung ultrasound can replace X-rays, for example, after chest surgery, when ionizing radiation is used as standard. After clinical testing of a new procedure using lung ultrasound, the need arose to automate the diagnostic procedure. A study by *Jaščur et al. (2021)* used DL in their work and created a new method that works with videos of lung ultrasound. The method consists of semantic segmentation of ultrasound images from the first images of the video. The lung region is exploited from which 2D images in the temporal dimension of the video are subsequently created, called M-mode images. The convolutional network model then classifies the presence or absence of lungsliding in a given time interval based on these images. In this work, they tested different parameters, and the best results were obtained for the 64-frame version with an accuracy of 89%, a sensitivity of 82% and a specificity of 92%.

A nice overview of works that deal with visual data such as 2D images, 3D images, and videos was provided in *Cazuguel (2017)* or *Singh, Sengupta & Lakshminarayanan (2020)*.

In recent years, transfer learning has made a significant progress in the medical domain. Transfer learning helped to address some of the problems related to this domain, such as data scarcity. Especially in medical image classification, such approaches are very well studied (*Kim et al., 2022*; *Mukhlif, Al-Khateeb & Mohammed, 2022*; *Hosseinzadeh Taher et al., 2021*). In medical video processing, there are also several studies available in which transfer learning is applied. For example, in *Klaiber et al. (2021)* the authors provide an extensive review of transfer learning applied on 3D convolutional networks, with some of the applications also from the medical domain. In *Aldahoul et al. (2021)* and *Lee et al. (2021)* the authors present particular approaches for transfer learning applied in the diagnosis of dysphagia using video frames and a pre-trained ResNet model for classification of laparoscopic videos. In our study, we focused on the explainability and interpretability aspects of the particular methods, therefore we did not include a more in-depth study of transfer learning applications.

## XAI for deep learning video processing in medical applications

Various uses of video processing with explanations in areas such as healthcare are also starting to come to the fore. Currently, various widely used technologies such as MRI, CT, or USG produce 3D images or short video sequences, which can be used by physicians to derive various diagnoses. To create systems that process these types of data, neural network architectures need to be modified, or the models used for 2D image data processing need to be combined with other methods. In this context, explainability also plays a role, as it can help developers create more accurate models and help physicians understand the behavior of the model and assess the accuracy of its predictions.

In their study, *Chittajallu et al. (2019)* present a human-in-the-loop XAI system for content-based image retrieval (CBIR) which they applied to video content from minimally invasive surgery (MIS) for surgical education. The method extracts semantic descriptors from MIS video frames using a self-supervised DL model. The model uses an iterative query refinement strategy, *i.e.,* based on user feedback, the model is repeatedly trained and refines the search results. The system receives a single frame from a video as input and tries to find similar frames and return them to the user. Finally, the XAI method creates saliency maps that provide visual explanations of the system's decisions. Based on the visual explanations, the user gives feedback to the system until the user is satisfied with the search result. Figure 6 is an example of their XAI system. The original (query) image is entered into the system and the content-based retrieval method is applied to the database of available images. The result containing similar images (bottom list of images) is visualized to the user and the system collects the user's feedback on search results. The feedback is guided by showing a heat map indicating the salient parts of a retrieved image that most influence its relevance/similarity to the query image. The human-in-the-loop approach is addressed by an iterative query refinement (IQR) strategy, where a binary classifier trained on the feedback is used to iteratively refine the search results.

*Manna, Bhattacharya & Pal (2021)* proposed SSLM, a self-supervised deep learning method for learning spatial context-invariant representations from MR (magnetic resonance) video frames. Video clips are used for the diagnosis of knee medical conditions. They used two models: the pretext model for learning meaningful spatial context-invariant representations and the downstream task model for class imbalanced multi-label classification. To analyze the reliability of their method, they show the gradient class activation mappings (Grad-CAM) for the detection of all classes. The salient regions are regions where the pretext model gains maximum information, which is then fed to the ConvLSTM model as a downstream task.

*Zhang, Wang & Lo (2021)* proposed a surgical gesture recognition approach with an explainable feature extraction process from minimally invasive surgery videos. They use Deep Convolutional Neural Network (DCNN) based on VGG architecture with the Grad-CAM XAI method. The class activation maps provide explainable results by showing the regions of the surgical images that strongly relate to the surgical gesture classification results. This work combines the DCNN network for spatial feature extraction and RNN for temporal feature extraction from surgery video.

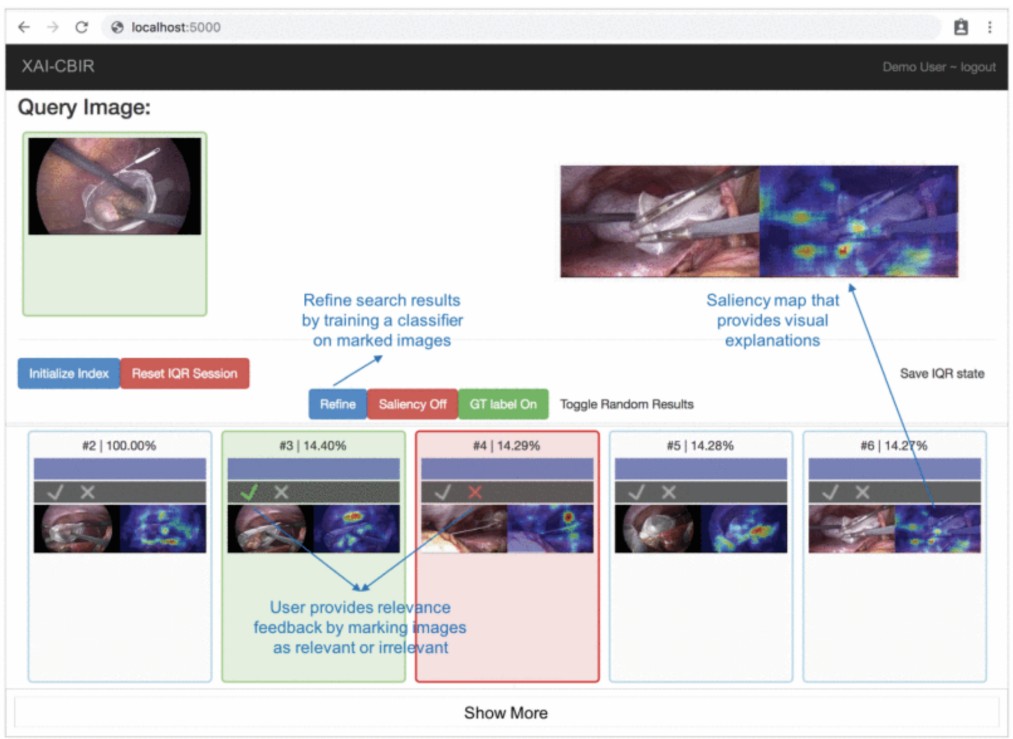

**Figure 6** Prototype of visual explanations from processing MIS video frames (*Chittajallu et al., 2019*).

*Knapi (2021)* present the potential of XAI methods for decision support in medical image analysis. They use three types of XAI methods on the same dataset to improve the comprehensibility of decisions provided by the CNN model. They use *in vivo* gastral images obtained by a video capsule endoscopy. In this study, they compare LIME, SHAP, and CIU methods, provide a questionnaire and quantitatively analyze it with three user groups with three distinct forms of explanations. Their findings suggest notable differences in human decision-making between various explanation support settings.

*Born et al. (2021b)* and *Born et al. (2021a)* proposed a publicly available lung POCUS dataset comprising samples from ultrasound videos of COVID-19 patients, pneumonia-infected lungs, and healthy patients. The dataset contains 247 videos recorded with either convex or linear probes. They proposed two models for the classification of lung ultrasound data, a frame-based model based on VGG-16 architecture and a video-based model based on 3D-CNN for 3D medical image analysis. They also used the Grad-CAM method for a frame-based model to explain model decisions for each target class. For example, CAMs highlight COVID-19 (highlighting a B-line), bacterial pneumonia (highlighting pleural consolidations), and healthy lungs (highlighting A-lines).

*Hughes (2020)* tried to explain optical flow models for video tasks. They proposed a method for trajectory-based explanations and used saliency maps to create red points to indicate current positions and green points to indicate history. They applied this method to the EchoNet-Dynamic dataset of videos of the heart.

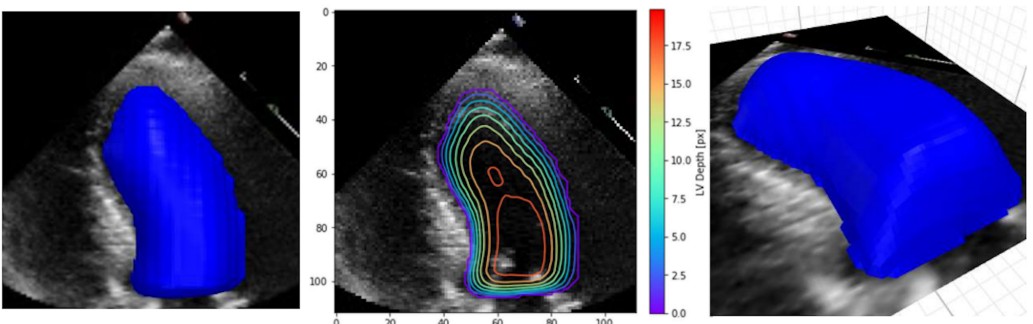

**Figure 7  Example depth map prediction shown in different perspectives and with contours to show geometry (*Duffy et al., 2021*).**

*Sakai et al. (2022)* proposed a novel deep learning XAI representation called graph chart diagram to support fetal cardiac of video ultrasound screening. They reduce the dimensionality of time-series information to the two-dimensional diagram using the TimeCluster method, which helps to find anomalies in long time-series. Therefore, they use autoencoders to compress dimensions. They proposed two techniques, view-proxy loss and a cascade graph encoder which improve performance and explainability by creating sub-graph chart diagrams of sets of substructures. In *Komatsu et al. (2021)* they proposed other techniques for explaining models of ultrasound images with bounding boxes of 18 anatomical substructures. They used these 18 classes to create a barcode-like timeline of video to highlight changes in the ultrasound video of the heart.

In their work, *Duffy et al. (2021)* have highlighted the lack of explainability and have re-examined explainable methods that fit the clinical workflow using 2D segmentation. However, they found out that the standard methods achieved lower accuracy. Therefore, they proposed the custom implementation of a DL model based on a frame-by-frame 3D depth-map approach that accounts for the standard clinical workflow while making the model explainable. This method is more applicable and can produce many predictions that clinicians can interpret easily and possibly improve the DL prediction. Figure 7 shows an example of their XAI approach.

*Fiaidhi, Mohammed & Zezos (2022)* used the XAI approach to provide better insights into DNN network decision-making in segmenting Ulcerative Colitis (UC) images. Their approach uses video processing methods such as summarization and automatic caption generation. In their model, they used the addition of contextual or heuristic information to increase the model's accuracy and better understand the model's decision-making. In their work, they investigated how adding heuristics for subtitles can increase the explainability of the model for UC severity classification. The model used a few video frames and classified them using a Siamese neural network. The output of the model, along with the captions from the gastroenterologist, were input to the LSTM network, which generated captions for the original video. However, the authors could not achieve an accuracy of descriptions higher than 62% due to the use of general embedding.

In their research work, *Acharya, Guda & Raovenkatajammalamadaka (2022)* used the transfer learning technique for the deep learning model to classify laparoscopic video pictures. They proposed eENetB0 and eENetB7 models based on the EfficientNet network and pre-trained on the ImageNet network. These models achieved 97.59% accuracy for eENetB0 and 98.78% for eENetB7 in the binary classification of video clips with blood and dry scenarios. GLENDA *Leibetseder et al. (2019)* dataset was used for training and testing models. The authors also provide a GUI application for real-time image-processing with human-like explanations of an area where the feature values are related to the model's prediction.

*Sakkos et al. (2021)* proposed a classification framework for infant body movements associated with the prediction of cerebral palsy from video data. Their novel method uses multiple deep learning approaches to classify the presence or absence of fidgety movements (FMs). Firstly they use OpenPose architecture to get the skeletal pose of the infant body. Specifically, to get trajectories of 8 selected body joints, including right and left hands, elbow, ankle, and knee. Each part of the body was processed separately by the LSTM network to find spatio-temporal motion in determining the abnormality of the body movement. Lastly, the CNN network processed the output of the LSTM network to classify the presence or absence of FMs. They also proposed the XAI method for the visualization of framework decisions. The framework provides a contribution score between 0-1 for each part of the body where higher values respond to a higher chance to present of FMs, and lower values correspond to a lower chance to FMs. There is also a visualization of the video split into 4 parts, where colors from purple to red are for the positive class, and color range from blue to green for the negative class. The authors claim their results correspond to a manual diagnostic tool such as general movement assessment (GMA).

Studies which deal with video processing using the above mentioned methods are summarized in Table 3.

Table 4 below summarizes the deep learning models used in the studies described. We observed the type of architecture used, the use of transfer learning, the performance of models, and the dataset type used in the studies.

Based on the presented survey of articles dealing with XAI deep learning models in medical video analysis we can summarize the following findings. In comparison with the traditional white box classification methods where suitable features need to be hand-crafted from the videos, models based on deep neural networks are able to extract the necessary features on their own. However, it is necessary to preprocess videos suitably. Most of the analyzed articles (8 out of 11) use frame-by-frame video processing, but there are also some other specific approaches, usually tightly connected with the concrete application specifics.

Regarding classification models used, the usually used DL architectures were successfully applied on 2D images with necessary adjustments or combinations of such architectures. In 4 out of 11 articles, transfer learning was used (in all cases model was pre-trained on ImageNet). The performance of the resulting models in terms of classification accuracy is usually very high, except for one very specific case and two articles where the classification performance was not documented.

**Table 3   XAI methods for video analysis.**

| Video processing type | Authors | Application | Model | XAI Methods | XAI evaluation method |
|---|---|---|---|---|---|
| Frame by frame | *Chittajallu et al. (2019)* | Human-in-the-loop XAI system for content-based image retrieval (CBIR) | ResNet50, IQR | Saliency maps | no XAI evaluation |
| Frame by frame | *Manna, Bhattacharya & Pal (2021)* | Self-supervised deep learning method for learning spatial context-invariant repre-setnations from MR (magnetic resonance) video frames (SSML) | SSLM, ConvLSTM | Grad-CAM | no XAI evaluation |
| Frame by frame | *Zhang, Wang & Lo (2021)* | Surgical gesture recognition approach with an explainable feature extraction process from minimally invasive surgery videos. | BML-indRNN, RNN + VGG16 | Grad-CAM | no XAI evaluation |
| Frame by frame | *Knapi (2021)* | Potential of XAI methods for decision support in medical image analysis - in vivo gastral images obtained by a video capsule endoscopy. | Custom CNN | LIME, SHAP, CIU | Human Evaluation User Study |
| Frame by frame | *Fiaidhi, Mohammed & Zezos (2022)* | Using XAI and heuristic information to increase model's performance on Ulcerative Colitis video data | Siamese neural network + LSTM | Caption heuristic | no XAI evaluation |
| Frame by frame | *Acharya, Guda & Raovenkatajammalamadaka (2022)* | Classification blood or dry scenarios of laparoscopic videos using EfficientNet and transfer learning | eENetB0, eENetB7 | Description based explanations of video | no XAI evaluation |
| Frame by frame | *Sakkos et al. (2021)* | Novel classification framework for infant body movements associated with prediction of cerebral palsy from video data | OpenPose + 1D CNN + LSTM | Contribution score and image highlights | no XAI evaluation |
| Frame-based classification + video-based | *Born et al. (2020)* | Lung POCUS dataset comprising samples from ultrasound videos and deep learning methods for the differential diagnosis of lung pathologies. | VGG16, VGG-CAM | CAMs (only for frame-based) | Evaluation by domain experts |
| Optical flow | *Hughes (2020)* | Explain optical flow models for video tasks. They proposed method for trajectory-based explanations and test on EchoNet-Dynamic dataset of videos of heart. | Optical Flow Decomposition | Trajectory-based explanations | Sanity check, Target Over Union, Target Over All |
| Barcode approach | *Sakai et al. (2022)* | Novel XAI representation called graph chart diagram, to support fetal cardiac of video ultrasound screening. | YOLOv2, auto-encoders | Custom - graph chart diagram | no XAI evaluation |
| 3D depth-map | *Duffy et al. (2021)* | DL model based on a frame-by-frame 3D depth-map approach that accounts for the standard clinical workflow. | DeepLabV3, ResNet | Custom | no XAI evaluation |

**Table 4   Deep learning models and video datasets.**

| Author | Model | Transfer learning | Model performance | Dataset |
|---|---|---|---|---|
| *Chittajallu et al. (2019)* | ResNet50, IQR | ImageNet pretrained | – | Public - Chochlec80 |
| *Manna, Bhattacharya & Pal (2021)* | SSLM, ConvLSTM | – | Accuracy 87.4% for abnormality class | Public - MRNet dataset |
| *Zhang, Wang & Lo (2021)* | BML-indRNN, RNN + VGG16 | ImageNet pretrained | Accuracy 87.1% | Public - JIGSAWS database |
| *Knapi (2021)* | Custom CNN | – | Accuracy 98.58% | Public - Red Lesion Endoscopy |
| *Fiaidhi, Mohammed & Zezos (2022)* | Siamese neural network + LSTM | – | Accuracy 62% | Public - KVASIR IBD data |
| *Acharya, Guda & Raovenkatajammalamadaka (2022)* | eENetB0, eENetB7 | Imagenet pretrained | Accuracy 98.78% (eEnetB7) | Public - GLENDA |
| *Sakkos et al. (2021)* | OpenPose + 1D CNN + LSTM | – | Accuracy 100% (MINI-RGBD), Accuracy 92% (RVI-25) | Public - MINI-RGBD, Not public - RVI-25 |
| *Born et al. (2020)* | VGG16, VGG-CAM | ImageNet pretrained (VGG16) | Accuracy 94% | Public - COVID-19 Lung ultrasound dataset |
| *Hughes (2020)* | Optical Flow Decomposition | – | – | Public - EchoNet-Dynamic |
| *Sakai et al. (2022)* | YOLOv2, auto-encoders | – | Accuracy 93.9% | Not public available |
| *Duffy et al. (2021)* | DeepLabV3, ResNet | – | $R2 = 0.82$ MAE = 4.05 | Public - EchoNet-Dynamic |

Analysis of XAI methods used for deep learning medical video classification showed that model-specific methods are dominating. From the methods presented in this article CAM and Grad-CAM, but authors developed also other, custom methods tightly connected with a specific type of applications, like contribution scores, trajectory-based explanations, or graph chart diagrams. In two articles we could find explanation methods providing some kind of textual descriptions. And only one article used model agnostic methods SHP, LIME and CIU described above.

Surprisingly, only three out of 11 analyzed articles provided some form of evaluation of the explanations provided by the used XAI method(s). In two cases human-grounded measures and in one computational measures were used.

## DISCUSSION

We think that the methodology used in this article provided sufficiently relevant, informative, and valuable insights into the rapidly evolving research domain of medical video analysis by means of XAI deep learning models. On the other hand, there may be some bias in case there exist also other relevant articles, which we missed because they could not be retrieved using the approach described at the beginning of this article. However, we think that the possible bias caused by this effect is very limited and does not threaten the validity of our findings. Another danger comes from the fact that this research area is evolving rapidly and new relevant articles may be published anytime.

New technologies that are non-invasive and becoming increasingly available can, in conjunction with artificial intelligence, help physicians to diagnose problems more quickly. One example is ultrasonography, which can effectively replace standard methods using ionizing radiation. For example, based on *Born et al. (2020)*, it is possible to classify COVID-19 patients using deep neural network applied to lung ultrasonography data. Another example is using the right diagnostic procedure to create an automated system for detecting a lung motion problem after thoracic surgery. The design of such a system was published in the article by *Jaščur et al. (2021)*. These (and many other) approaches achieve interesting results, but suffer from a lack of explainability, which is required in healthcare, both by physicians and legislation. Using more transparent models or explainable methods can help explain AI decisions. In turn, choosing an appropriate architecture can help to improve the model prediction. For example, using 3D features that can be extracted from the video can improve prediction and simplify the application of explainability (*Duffy et al., 2021*).

USG is one of the most common medical imaging techniques. It has several advantages over other techniques such as X-ray, CT, and MRI. USG does not use ionizing radiation and is portable, and cost-effective (*Liu et al., 2019*). However, the disadvantage of USG is the low quality of imaging due to low resolution and noise. The observation's content depends on the physician's experience and the hardware specification of the equipment. Existing approaches using DL methods on USG data mainly deal with classification, detection, segmentation, and registration tasks. The tasks include analyzing distinct anatomical structures such as the heart, muscle, breast, liver, lung, etc.

In classification tasks on lung USG, AI classifies the presence or absence of pathological features from images, mainly using 2D CNN architecture. These architectures are sufficient in case of static features like tumors and lesions in the breast and liver. The problem occurs if we use 2D architecture to analyze movement patterns in biomedical images, such as the presence of lung sliding. We need to use a 3D CNN architecture to capture motion over time. However, such an architecture tends to be more demanding on system resources and training time, and it is more challenging to implement the explainability of such a complex architecture.

However, as we presented in this article, there are similar open problems with the explainability of the video analytical methods, yet to be solved present in other domains than medicince. The most important open issues will be summarized in the following subsection.

## Open issues and future trends

As the application of XAI approaches in video processing tasks in the medical domain remains a very active research topic, there are several open problems to be solved in the future. One such problem lies in the lack of a qualitative metric for explanations. Nowadays, the most common approach in the medical domain, is getting feedback directly from the domain expert (clinician) expertise *e.g.*, using a questionnaire. This approach has two major downsides. Firstly, it is time consuming and when handling multiple data sources it can be difficult to achieve in real-world deployment. Then, in the case of visual image/video explanations, there is subjectivity in such an approach, as experts opinions on the provided explanations may be biased. Therefore, the need for fully-automated evaluation of explanations (*e.g.*, using some objective metric) still remains among the open problems yet to be solved. Besides the evaluation, there are several issues related to the availability and quality of the training data. In the medical domain, the availability of the data is a complicated issue. Medical data are very sensitive, as they represent a portion of a person's private patient's data. Collection and storage of such data must involve actions to ensure the trust and security aspects. Then, there is the aspect of obtaining the class labels (as the majority of the analytical tasks are supervised). Labeling is mostly being done manually by the experts themselves, which is very time-consuming and resource-demanding. Also, in manual annotation, the subjectivity of the expert opinion may influence the correctness of the data labeling. One of the consequences of these factors is that there are not many available training datasets and those available are rather small. To overcome these problems, a combination of existing approaches can be adopted. For example, augmentation techniques can be used to enhance the volume of the datasets, as these approaches have proven to be effective in image and video processing tasks from other domains. Other techniques, such as transfer learning or self-supervised learning may help with the labeling, but must be further explored and evaluated on medical data.

## CONCLUSION

This article summarized and reviewed the current approaches to explainability techniques applied to deep learning models for medical video analysis. We started by introducing the fundamental terminology in the area of explainability and interpretability, focusing more on its importance in the healthcare domain. We summarized the requirements for an explainable AI system deployed in real-world applications and summarized the desiderata for XAI in this domain. Then, we provided an overview of classical XAI methods which can be used in video analytical tasks. After this, we reviewed the works focused on explaining the decision process of deep learning applied to medical video analysis. Here, we analyzed the existing approaches to medical video analysis and EAX techniques applied in this area. Some of the approaches utilize similar methods to those that are applied to medical imaging, but adapted with dynamic aspects to address the specifics of video data. We also highlighted open research issues in this area, some of them being similar and related to explainability issues in medical image analysis. This particular area is not currently as heavily studied as other tasks, therefore we think that providing a review of the currently used approaches may be beneficial for the research community focusing on this field.

### Funding

This work was supported by the Slovak Research and Development Agency under the contract No. APVV-20-0232 and contract No. APVV-17-0550 and by the Slovak VEGA research grant No. 1/0685/21. The funders had no role in study design, data collection and analysis, decision to publish, or preparation of the manuscript.

### Grant Disclosures

The following grant information was disclosed by the authors:
The Slovak Research and Development Agency: APVV-20-0232, APVV-17-0550.
The Slovak VEGA research: 1/0685/21.

### Competing Interests

The authors declare there are no competing interests.

### Author Contributions

- Michal Kolarik conceived and designed the experiments, performed the experiments, analyzed the data, prepared figures and/or tables, and approved the final draft.
- Martin Sarnovsky conceived and designed the experiments, performed the experiments, analyzed the data, prepared figures and/or tables, and approved the final draft.
- Jan Paralic conceived and designed the experiments, prepared figures and/or tables, authored or reviewed drafts of the article, and approved the final draft.
- Frantisek Babic conceived and designed the experiments, prepared figures and/or tables, authored or reviewed drafts of the article, and approved the final draft.

## Data Availability

This is a literature review.

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
