# Peer review of "Explainability of deep learning models in medical video analysis: a survey"

_PeerJ Computer Science, doi:10.7717/peerj-cs.1253_

## Round 0.1 · original submission · Minor Revisions

I invite you to submit a revised manuscript addressing all reviewers' comments. Also, respond to the following concerns:
1. the survey methodology needs to be better described. Clarify if it is a systematic literature review, a bibliometric analysis, or a meta-analysis. How did you select the articles? A PRISMA Diagram is essential to support the methodology.
2. Correct all grammatical/typo errors.

·

Basic reporting

This is a review paper addressing a very important issue of explainability of Deep Learning models for Medical Video Processing.

This paper is a review paper.
This paper provides a review of the current approaches and applications of explainable deep learning for a specific area of medical data analysis.

Experimental design

Experimental designs used by other researchers, equations of Analytical methods are given in depth and discussed in detail.

Validity of the findings

Validity of Findings: They have reviewed around 50 papers and in the end given future directions to researchers. They have included results of other researchers with appropriate citations.
This paper is very useful for researchers working in Medical Video Processing domain.

Additional comments

The paper address a very important aspect of AI systems esp. when used for medical diagnosis: that of exlpainability. The paper has presented in depth multiple ways its can be done. It present a comprehensive study.

Conclusion: This paper has discussed multiple algorithms and recent methods used for medical Video Processing. If the journal accepts review papers, this paper definitely consolidates the available methods in the fore-mentioned domain and is useful across multiple disciplines.

·

Basic reporting

The paper presents an overview of existing methods, evaluation metrics, and explainability methods for processing of the video data in the medical domain. The survey is broad, and the discussion on background is sufficient and enough references are discussed. The paper, however, needs to be improved according to the comments before it could be considered for publication.

Experimental design

1. Improve the description of the search methodology and present more details including the bibliographic databases and search queries used to find the articles.
2. Present the summary concepts (principles) as a taxonomy (ontology).
3. Summarize the advantages/disadvantages of discussed methods in a table.
4. Present a more suitable (such as disease-related) example of Grad-CAM visualization in Figure 2 and activity recognition in Figure 3.

Validity of the findings

Discuss the limitations of your methodology such as bias, and threats to the validity of your findings.

Additional comments

Improve the language. There are many spelling errors and typos such as “Comparsion”.

Reviewer 3 ·

Basic reporting

No comment

Experimental design

No comment

Validity of the findings

No comment

Additional comments

This paper is well organized and it explains the relevant techniques in detail.

·

Basic reporting

This is an interesting work which introduces XAI methods for medical video analysis, focusing on pre-processing, segmentation, data preparation, and post-processing. Large-scale retrieval increases video processing efficiency. Paper is well written and organized however the manuscript requires several major revisions before considering for publication.

Experimental design

1. Authors have claimed in the introduction section that XAI addresses the issues of video pre-processing, classification, choice of model selection and low performance measure. However, the results in view of these parameters are not explained clearly in the paper.
2. Fig 4 is not clear authors should clearly explain the efficacies of diagram.
3. The author should add more recant papers.
4. No discussion on the usage of deep transfer learning in medical video analysis. Since medical data are sparse in nature which justifies the efficacy of transfer learning in medical domain.

Validity of the findings

5. More comparative analysis should be carried out to strengthen the contribution of the paper. The datasets can also be discussed with some XAI models and their comparative performances. Authors are strongly recommended to present the state of comparison between various models and available public dataset.
6. The language seems to be unprofessional at many places. Monotonous lines been used throughout the whole paper.

Additional comments

The work is also close to the aims and scope of the journal and so, I think it can be accepted for publication in its revised version. Prior to publication it is important to further improve the use of English language in the manuscript.

---

## Round 0.2 · Minor Revisions

Thank you for addressing the reviewer concerns. The manuscript appears to be ready for publication with just a couple of comments.

In the case of Figure 5, it could be helpful to the reader to have the description of the different methods and their corresponding output included within the figure caption itself (as well as the main text).

Several figures are "adapted from" a provided reference. Are these "fully" adapted or might they require permission from the original publisher?

Thank you for clarifying the above and your submission.

·

Basic reporting

The manuscript has been well revised. The authors have addressed all my suggestions and comments. I have no further comments, and recommend the manuscript to be accepted for publication.

Experimental design

The methodology and design of the study is good.

Validity of the findings

The findings of this review are valid.

---

## Round 0.3 · accepted · Accept

Thank you for addressing the concerns. Congratulations again.